# Mono-mix strategy enables comparative proteomics of a cross-kingdom microbial symbiosis

Sunnyjoy Dupuis[1,2], Usha F. Lingappa[2], Samuel O. Purvine[3], Lauren Chiang[1], Sean D. Gallaher[2], Carrie D. Nicora[3], Mary S. Lipton[3], Sabeeha S. Merchant[1,2,4,5]*

**1** Department of Plant and Microbial Biology, University of California, Berkeley, California, United States of America, **2** California Institute for Quantitative Biosciences, University of California, Berkeley, California, United States of America, **3** Earth and Biological Sciences Division, Pacific Northwest National Laboratory, Richland, Washington, United States of America, **4** Environmental Genomics and Systems Biology Division, Lawrence Berkeley National Laboratory, Berkeley, California, United States of America, **5** Department of Molecular and Cell Biology, University of California, Berkeley, California, United States of America

\* sabeeha@berkeley.edu

## Abstract

Cross-kingdom microbial symbioses, such as those between algae and bacteria, are key players in biogeochemical cycles. The molecular changes during initiation and establishment of symbiosis are of great interest, but quantitatively monitoring such changes can be challenging, particularly when the microorganisms differ greatly in size or are intimately associated. Here, we analyze output from label-free, data-dependent acquisition (DDA) LC-MS/MS proteomics experiments investigating the well-studied interaction between the alga *Chlamydomonas reinhardtii* and the heterotrophic bacterium *Mesorhizobium japonicum*. We found that detection of bacterial proteins decreased in coculture by 50% proteome-wide due to the abundance of algal proteins. As a result, standard differential expression analysis led to numerous false-positive reports of significantly downregulated proteins, where it was not possible to distinguish meaningful biological responses to symbiosis from artifacts of the reduced protein detection in coculture relative to monoculture. We show that data normalization alone does not eliminate the impact of altered detection on differential expression analysis of the cross-kingdom symbiosis. We assessed two additional strategies to overcome this methodological artifact inherent to DDA proteomics. In the first, we combined algal and bacterial monocultures at a relative abundance that mimicked the coculture, creating a "mono-mix" control to which the coculture could be compared. This approach enabled comparable detection of bacterial proteins in the coculture and the monoculture control. In the second strategy, we enhanced detection of lowly abundant bacterial proteins by using sample fractionation upstream of LC-MS/MS analysis. When these simple approaches were combined, they allowed for meaningful comparisons of nearly 10,000 algal proteins and over 4,000 bacterial

**Data availability statement:** All raw proteomics data files are available from https://massive.ucsd.edu/ under the accession MSV000098360. All processed proteomics data and other relevant data are within the manuscript and its Supporting information files.

**Funding:** This work was supported by The Gordon and Betty Moore Foundation Symbiosis in Aquatic Systems Initiative Investigator Award GBMF9203 to S.S.M. (https://doi.org/10.37807/GBMF9203). Proteomic analysis was performed on a Facilities Integrating Collaborations for User Science (FICUS) project award (10.46936/fics.proj.2021.60010/60000390) from the Environmental Molecular Sciences Laboratory, a Department of Energy (DOE) Office of Science User Facility sponsored by the Office of Biological and Environmental Research and located at Pacific Northwest National Laboratory (PNNL). PNNL is operated by Battelle for the DOE under Contract No. DE-AC05-76RL01830. S.D. acknowledges support from the National Institutes of Health T32 Genetic Dissection of Cells and Organisms training grant 1T32GM132022-01. U.F.L. acknowledges support from the μBiospheres Scientific Focus Area grant SCW1039 supported by the Genome Sciences Program of the U.S. Department of Energy's Office of Biological and Environmental Research. The funders had no role in study design, data collection and analysis, decision to publish, or preparation of the manuscript. There was no additional external funding received for this study.

**Competing interests:** The authors have declared that no competing interests exist.

proteins in response to symbiosis by DDA. They successfully recovered expected changes in the bacterial proteome in response to algal coculture, including upregulation of sugar-binding proteins and transporters. They also revealed novel proteomic responses to coculture that guide hypotheses about algal-bacterial interactions.

## Introduction

Cross-kingdom microbial symbioses abound in nature. Interactions between bacteria, archaea, fungi, and protists provide important ecosystem services, shape evolution, and drive biogeochemical cycling. For instance, syntrophic bacteria and archaea of the rumen degrade recalcitrant plant biomass in the ruminant digestive tract and constitute a major source of methane emissions globally [1,2]. Mycorrhizal fungi and symbiotic bacteria in their "hyphosphere" provide nitrogen, phosphorus, and other minerals to plants in exchange for carbon, increasing plant productivity [3–5]. Lichenized photobionts release glucose to symbiotic fungi and are protected by fungal structures and metabolites [6]. Phytoplankton are responsible for much of the Earth's primary productivity, serving as a major source of organic carbon for marine food webs and providing a habitat for heterotrophic bacteria in the "phycosphere" [7,8]. In each instance, tight association of the symbionts, their divergent structural characteristics, and large differences in their abundances and metabolic rates can pose challenges to studying the microorganisms' molecular responses to one another.

Cross-kingdom algal-bacterial interactions are an important component of global carbon cycling [9–11]. Among the most well studied algae is the unicellular green alga *Chlamydomonas reinhardtii*. This alga is experimentally tractable, can perform both photosynthesis and heterotrophy, and exhibits rhythms in growth and metabolism over diurnal cycles [12,13]. In recent decades, studies have begun to elucidate how *C. reinhardtii* interacts with various bacteria [14–23]. For example, the interaction between the alga and the vitamin $B_{12}$-producing α-proteobacterium *Mesorhizobium japonicum* has been well documented [15,22,24–26]. This bacterium receives a small but significant amount of photosynthetically fixed carbon in coculture, primarily via algal cell lysis [26]. Yet, it is unknown what specific alga-derived substrates *M. japonicum* and other bacteria may receive from *C. reinhardtii*, and how such partners sense and respond to one another.

Quantitative investigations of algal-bacterial interactions can be challenging due to technical barriers for studying eukaryotic and prokaryotic microbes simultaneously without disrupting the interaction, particularly when the partners differ in abundance and size by orders of magnitude [27]. Algal cells are typically much larger than prokaryotic cells. For example, *C. reinhardtii* cell volume ranges from roughly 70–3000 $\mu m^3$, whereas *M. japonicum*'s cell volume is around 1–5 $\mu m^3$ [26]. Although this bacterium can reach up to $10^9$ cells/ml in coculture, 100-fold higher than the maximum algal cell density, *M. japonicum* makes up only 10% of the total biomass in mixed cultures, at best. Accurate quantification of bacterial cell density itself is challenging in algal cocultures: optical density cannot be used, as algal pigments influence absorbance, so cells

must be counted or estimated from colony forming units (CFU). In addition, lowly abundant bacterial molecules (e.g., proteins) are difficult to detect quantitatively amidst an abundance of algal molecules.

Liquid chromatography and tandem mass spectrometry (LC-MS/MS) proteomics is a useful tool for studying microbial interactions [28–31]. The most widely used methodology is label-free, data-dependent acquisition (DDA). In this "shotgun" approach, peptide precursor ions are chosen stochastically for fragmentation from an MS scan. Precursor ion selection is influenced by the complexity of the analyte, and ions are more likely to be chosen by the mass spectrometer if they are highly abundant at the expense of less abundant ions [32,33]. Therefore, although eukaryotic and prokaryotic peptides present in a sample are generally both detectable by MS with respect to their chemical properties, lowly abundant bacterial peptides may be missed in coculture samples that are dominated by algal peptides. Systematic differences in detection related to differences in analyte complexity complicate differential expression analysis, particularly between coculture and monoculture samples [34–36]. Such comparisons are of great interest, as they have the potential to reveal proteomic responses to initiation and establishment of symbiosis. This issue can be avoided by first separating the symbiotic species from one another (e.g., through differential centrifugation) and then analyzing enriched sample fractions by DDA proteomics [37,38]. However, disrupting microbial interactions in this way is less desirable, as separation procedures may change gene expression, and as enriched samples may be biased towards cells that are more loosely associated with their symbionts.

Several studies have reported attempts to capture proteomic changes during algal-bacterial interactions using DDA proteomics of mixed culture samples. Proteomic analysis of cocultures of the alga *Lobomonas rostrata* with *M. japonicum* using the iTRAQ labelling approach coupled with DDA yielded identification of only 4 bacterial proteins (in addition to 588 algal proteins) [39]. In a study of the interaction between *C. reinhardtii* and the growth promoting bacterium *Arthrobacter* strain P2b, many fewer bacterial proteins were detected in samples collected from coculture than from bacterial monocultures during label-free DDA proteomic analysis [21]. In another label-free DDA study investigating symbiosis between *C. reinhardtii* and the algicidal bacterium *Paenibacillus polymyxa*, hundreds of bacterial proteins were identified, many of which were significantly altered in abundance in the presence of the alga, but successful detection may have only been possible because of the minimal algal biomass resulting from algicidal activity [40].

Here, we quantitatively assess label-free DDA proteomics of *C. reinhardtii* and *M. japonicum* from mixed cultures. We show that the presence of algal proteins leads to a 50% reduction in bacterial protein detection in a typical shotgun proteomics experiment of cocultures and monocultures. These global differences in detection resulted in false positives during typical differential expression analyses, complicating identification of meaningful changes to the bacterial proteome in response to symbiosis. We explored three accessible strategies to improve DDA proteomics analysis of lowly abundant bacterial proteins from mixed *C. reinhardtii-M. japonicum* cultures: data normalization, the use of a "mono-mix" control, and sample fractionation. Together, these accessible strategies allowed us to confidently determine significant changes in bacterial protein abundance in coculture when it is relying on an alga for its growth. We discuss limitations and propose additional improvements to enable robust comparative proteomics of microbial symbioses.

## Materials and methods

### Strains and culture conditions

The *C. reinhardtii* strain closely related to strain S24-, which is often studied with *M. japonicum*, was used in the first experiment (Sample Set 1) under continuous light [26]. The *C. reinhardtii* cell-wall reduced strain CC-5390 was used in the experiments comparing continuous and diurnal light (Sample Set 2). This strain is amenable to synchronization under diurnal cycles, yielding populations of cells exhibiting nearly the same physiological state and gene expression, which provide a system with exceptional signal-to-noise for resolving diurnal patterns [41]. *M. japonicum* strain MAFF303099 was used in all experiments.

Cultures were grown in amended High Salt minimal medium (HSM) [26,42], which contains Kropat's trace element solution [43], plus 2 mM $MgSO_4 \cdot 7H_2O$, 10 μg/l biotin, and 10 μM $CoCl_2$ to support bacterial growth, but lacks a reduced carbon source. Cultures were grown in 250 ml Erlenmeyer flasks containing 100 ml medium and bubbled with filter-sterilized air (provided by an aquarium pump). Algal cultures and cocultures were incubated at 28°C, agitated at 110–130 rpm, and illuminated with 200 μmol photons/m$^2$/s cool white light (Sylvania Dulux L55WFT55DL/841) continuously or under diurnal 12-h-light/12-h-dark cycles.

*C. reinhardtii* cells were precultivated axenically in photoautotrophic conditions. For the experiment comparing continuous and diurnal light, algal precultivation occurred in 400 ml flat-panel photobioreactors (Photon System Instruments, Drásov, Czechia) as previously described under the respective light regime to achieve adequate synchrony of the diurnal populations [41]. Bacterial cells for all experiments were precultivated in 14 ml polystyrene round-bottom tubes with 3 ml amended HSM containing 0.2% sucrose, agitated at 200 rpm, and incubated at 28°C. Bacterial cells were collected by centrifugation at 8000 x*g* and washed three times in 0.85% NaCl to remove sucrose before use as inoculum. Optical density at 600 nm ($OD_{600}$) was used to estimate the cell density of washed cell suspensions used for inoculation. Biological replicates refer to replicate flasks inoculated and incubated together in an experiment.

## Estimation of cell density

*C. reinhardtii* cell density was determined using a Z2 Coulter Particle Count and Size Analyzer (Beckman Coulter, CA, USA). *M. japonicum* cell density was estimated as CFU determined from serial dilutions of culture samples on Tryptone Yeast (TY) agar plates after 4 d of incubation at 30°C in the dark.

## *In silico* assessment of protein physicochemical characteristics

The physicochemical properties of proteins encoded by *C. reinhardtii* and *M. japonicum* were assessed according to Shi *et al.* 2024 [36]. The *C. reinhardtii* reference genome assembly and annotations v6.1 protein FASTA file from was downloaded from https://phytozome-next.jgi.doe.gov [44]. The *M. japonicum* MAFF303099 RefSeq protein FASTA file was downloaded from https://www.ncbi.nlm.nih.gov/data-hub/taxonomy/266835/. FASTA files were read into R using the "seqinr" package (v4.2-36). The isoelectric point (pI) of each protein was estimated using the pI function in the R package "Peptides" (v2.4.6) using pKscale = "EMBOSS". The grand average of hydropathy (GRAVY) hydrophobicity score of each protein was estimated using the hydrophobicity function from the R package "Peptides" (v2.4.6) using scale = "KyteDoolittle". The molecular weight of each protein was determined using the mw function from the R package "Peptides" (v2.4.6) using monoisotopic = FALSE, avgScale = "expasy", label = "none", and aaShift = NULL.

## Proteomic sample collection and digestion

40 ml of culture was collected either 24, 36, or 48 h after inoculation. Cell pellets were collected by centrifugation at 8000 x*g* for 2 min at 4°C and were resuspended in 10 mM Na-PO$_4$ buffer (pH 7.0). Cell pellets were again collected by centrifugation at 8000 x*g* for 1 min at 4°C, snap frozen in liquid N$_2$, and stored at –80°C until further processing.

Cell pellets were diluted in 300 μl of 8 M urea and cells were lysed by bead beating in Micro-Organism Lysing Mix glass bead tubes for 45 s at 5.5 m/s using a Bead Ruptor Elite bead mill homogenizer (OMNI International, Kennesaw, GA). Lysate was collected by centrifugation at 1,000 x*g* for 10 min at 4°C. Protein concentration was determined by bicinchoninic acid (BCA) assay (Thermo Scientific, Waltham, MA USA). Then, 10 mM dithiothreitol (DTT) was added and samples were incubated at 60°C for 30 min with constant shaking at 800 rpm. 40 mM iodoacetamide (IAA) was then added and samples were incubated at ambient temperature in the dark for 30 min.

Protein samples were then digested with sequencing grade trypsin (United States Biological, Salem, MA) at a 1:50 (w/w) trypsin-to-protein ratio for 3 h at 37°C in the presence of 100 mM NH$_4$HCO$_3$ and 1 mM CaCl$_2$. Digested samples were desalted using a 4-probe positive pressure Gilson GX-274 ASPEC™ system (Gilson Inc., Middleton, WI) with

Discovery C18 50 mg/1 ml solid phase extraction tubes (Supelco, St. Louis, MO), using the following protocol: 3 ml of methanol was added for conditioning followed by 3 ml of 0.1% trifluoroacetic acid (TFA) in $H_2O$. The samples were then loaded onto the column followed by 4 ml of 95:5 $H_2O$:acetonitrile (ACN), 0.1% TFA. Samples were eluted with 1 ml 80:20 ACN:$H_2O$, 0.1% TFA. The solvent was completely removed in a vacuum concentrator (Labconco, Kansas City, MO) and the peptides subsequently reconstituted in 100 µl pure water. Peptide concentration of the final mixture was determined using a BCA assay, and samples were diluted to 0.10 µg/µl with pure water for global proteomics LC-MS/MS analysis.

### Reversed phase fractionation of peptides

Peptide mixtures were separated into 6, 12, or 24 fractions by high-resolution reversed phase UPLC using a nanoAC-QUITY UPLC® system (Waters Corporation, Milford, MA) equipped with an autosampler. Capillary columns, 200 µm i.d. x 65 cm long, were packed with Jupiter 3.0 µm C18 300 Å bonded particles (Phenomenex, Torrance, CA). Separations were performed at a flow rate of 2.2 µl/min on binary pump systems, using 10 mM ammonium formate (pH 7.5) as mobile phase A and 100% ACN as mobile phase B. 48 µl of the peptide mixtures (0.25 µg/µl) were loaded onto the column and separated using a binary gradient of 1% B for 35 min, 1–10% B in 2 min, 10–15% B in 5 min, and 15–25% B in 35 min, 25–32% B in 25 min, 32–40% B in 13 min, 40–90% B in 43 min, held at 90% B for 2 min (column washed and equilibrated from 90–50% B in 2 min, 50–95% B in 2 min, held at 95% B for 2 min and 95–1% in 4 min). The capillary column eluent was automatically deposited every minute into 12 mm x 32 mm polypropylene vials (Waters Corporation, Milford, MA) starting at minute 60 and ending at minute 170, over the course of the 180 min LC run. Fractions were concatenated into 6, 12, or 24 vials by collecting fractions from vial 1 to vial 6–24 and then returning to vial 1 (and so on). Prior to peptide fraction collection, 100 µl of water was added to each vial to facilitate the recovery of small droplets added into the vial. Each vial was completely dried in a vacuum concentrator (Labconco, Kansas City, MO) and reconstituted in 20 µl 2% ACN, 0.1% formic acid.

### LC-MS/MS proteomics acquisition

A nanoACQUITY dual pumping UPLC system (Waters Corporation, Milford, MA) was configured for on-line trapping of a 5 µl injection at 5 µl/min for 10 min followed by reverse-flow elution onto the analytical column at 200 nl/min. The trapping column was slurry packed in-house using 360 µm o.d. x 150 µm i.d. fused silica (Polymicro Technologies Inc., Phoenix, AZ) Jupiter 5µm C18 media (Phenomenex, Torrence, CA) with 2 mm sol-gel frits on either end. The analytical column was slurry packed in-house using Waters BEH 1.7 µm particles packed into a 35 cm long, 360 µm o.d. x 75 µm i.d. column with an integrated emitter (New Objective, Inc., Littleton, MA). Mobile phases consisted of (A) 0.1% formic acid in water and (B) 0.1% formic acid in ACN with the following gradient profile (min, %B): 0, 1; 10, 8; 105, 25; 115, 35; 120, 75; 123, 95; 129, 95; 130, 50; 132, 95; 138, 95; 140, 1.

MS analysis was performed using a Thermo Eclipse mass spectrometer (Thermo Scientific, San Jose, CA). The ion transfer tube temperature and spray voltage were 300 ºC and 2.4 kV, respectively. Data were collected for 120 min following a 27 min delay from when the trapping column was switched in-line with the analytical column. FAIMS was used at CVs –40 V, –60 V, and –80 V. FT-MS spectra were acquired from 300–1800 m/z at a resolution of 120 k (AGC target 4e5) and while the top 12 FT-HCD-MS/MS spectra were acquired in data-dependent mode with an isolation window of 0.7 m/z and at an orbitrap resolution of 50 k (AGC target 5e4) using a fixed collision energy (HCD) of 32 and a 30 s exclusion time.

### LC-MS/MS proteomics data processing

All LC-MS/MS datafiles were converted to mzML using MSConvert with all peaks kept during centroiding [45]. MZRefinery was used to calibrate search mass values from the MS2 data, which were then input to MS-GF+ [46]. All data were searched against a single concatenated FASTA file containing amino acid sequences from both the *C. reinhardtii*

reference genome assembly and annotations v6.1 (https://phytozome-next.jgi.doe.gov/info/CreinhardtiiCC_4532_v6_1) (32,670 protein entries, 17,691 protein-coding genes) [44], the *M. japonicum* MAFF303099 genome assembly and annotations (https://www.ncbi.nlm.nih.gov/datasets/genome/GCF_000009625.1/) (7,105 protein entries, 7,162 protein-coding genes), and commonly observed contaminant proteins (16 protein entries). We employed a target/decoy approach with +/–20 ppm parent mass tolerance, dynamic modification of oxidized methionine, and allowance of partially tryptic peptide candidates [47]. MS abundances were extracted using StatsMomentsArea via the MASIC software to provide area-under-the-elution-curve values (i.e., extracted ion chromatograms) [48]. Data were collated using the MAGE software suite (https://github.com/PNNL-Comp-Mass-Spec/Mage), imported into SQL Server, and cross-checked for completeness. FDR was controlled to 1% (q-value ≤ 0.010297 gave 16,354 decoy identifications out of 1,635,330 total for Sample Set 1; q-value ≤ 0.0103619 gave 31,592 decoy identifications out of 3,134,149 total for Sample Set 2). Peptide-to-protein relationships were subjected to parsimony such that a protein was assigned if a) the peptide was uniquely mapped to that protein, b) was non-uniquely mapped to only one protein with a different uniquely mapping peptide, c) was non-uniquely mapped to a protein more other non-uniquely mapping peptides than any others, or d) was mapped to the first occurrence of the peptide in the search FASTA file when more than one protein had similar maximal peptide counts.

Data from samples that had been fractionated were then processed separately from data from samples that had not been fractionated. To infer protein abundance from peptide abundance, the simple roll-up method was applied, in which peptide abundances were $\log_2$ transformed, mean central tendency normalized, then grouped by their gene ID and summed per sample, resulting in protein abundances as $\log_2$-transformed MASIC values. This common approach was preferred here over other existing approaches for label-free protein quantification (e.g., MaxLFQ [49]), as it is computationally simple, uses all peptide information available, inherently weights signals based on their abundances (i.e., more abundant peptides influence protein abundance determination more than less abundant peptides whose signal may be noisier), and is suitable for LC-MS-level quantitation [50].

Only proteins with at least two unique peptides for quantitation that were detected in at least two of the three biological replicates in any condition were analyzed. In the unfractionated pilot experiment (Sample Set 1), this included 9871 proteins (6113 *C. reinhardtii* proteins, 3758 *M. japonicum* proteins). In the unfractionated mono-mix experiment (Sample Set 2), this included 9784 proteins (7039 *C. reinhardtii* proteins, 2745 *M. japonicum* proteins). In the mono-mix experiment comparing protein detection before and after peptide sample fractionation (Sample Set 2, fractionated), this included 15,464 proteins (10,858 *C. reinhardtii* proteins, 4606 *M. japonicum* proteins). Missing values were handled by exclusion rather than imputation.

$\log_2$-transformed protein MASIC values were quantile normalized using the normalize.quantiles function from the R package *preprocessCore* (v1.66.0) [51,52]. Quantile normalization was performed independently for each organism: the distributions of *C. reinhardtii* protein abundances across the relevant samples were quantile normalized together, and the distributions of *M. japonicum* protein abundances across the relevant samples were quantile normalized together. Normalized and unnormalized abundances for proteins analyzed in the unfractionated pilot experiment (Sample Set 1) are provided in S2 Table. Normalized and unnormalized abundances for the proteins analyzed in the unfractionated mono-mix experiment (Sample Set 2) are provided in S3 Table. Normalized and unnormalized abundances for the proteins analyzed in the mono-mix experiment before and after peptide sample fractionation (Sample Set 2, fractionated) are provided in S5 Table.

The average protein abundances were calculated after excluding missing values and the $\log_2$-fold change in protein abundance was calculated as (mean abundance in coculture) – (mean abundance in monoculture). To calculate the Z-score abundance for each protein in a given experiment, missing values were excluded, then the mean and standard deviation of quantile-normalized MASIC values for each protein across the experimental conditions was calculated, and the Z-score was calculated as (mean norm. MASIC – (mean of all mean norm. MASIC))/(sd of all mean norm. MASIC).

## Clustering analyses

*k*-means clustering was performed on the Z-score of quantile-normalized mean protein abundances in the unfractionated mono-mix experiment (Sample Set 2), separately for *C. reinhardtii* proteins and *M. japonicum* proteins that were detected in at least two biological replicates in all conditions (4737 *C. reinhardtii* proteins, 828 *M. japonicum* proteins). The fviz_nbclust function from the R package *factoextract* (v1.0.7) was used to generate elbow plots to determine the optimal *k* for clustering. The kmeans function from the R package *stats* (v4.4.0) was then applied using iter.max = 1000, centers = 6 for *C. reinhardtii* proteins or centers = 3 for *M. japonicum* proteins. Cluster assignment is provided in S3B Table. Heatmaps were generated using the pheatmap function from the R package *pheatmap* (v1.0.12).

Principal component analysis (PCA) was performed on quantile-normalized protein abundances of individual replicate samples in the unfractionated mono-mix experiment. This was done separately for *C. reinhardtii* proteins and *M. japonicum* proteins that were detected above the limit of quantitation (two standard deviations below the mean MASIC value in the experiment) in all samples (all replicates in all conditions: 3817 *C. reinhardtii* proteins, 569 *M. japonicum* proteins). The pca function was applied from the R package *PCAtools* (v2.16.0) and the results were visualized using the biplot function.

## Statistical tests

Significant differences in culture densities and the number of proteins detected in different sample groups were assessed by two-tailed Student's t-tests. Significant differences in the distribution of protein abundances (only for those proteins that were detected in at least two of the three biological replicates in both the monoculture control condition and the respective coculture condition) were assessed by Wilcoxon rank-sum tests (Mann-Whitney tests) using the wilcox.test function in the R package *stats* (v4.4.1).

Significant differences in the abundance of a given protein in coculture relative to monoculture were assessed using the t.test function in the R package *stats* (v4.4.0). Only proteins that were detected in at least two of the three biological replicates in both the monoculture control condition and the respective coculture condition were analyzed. In the unfractionated pilot experiment (Sample Set 1), this included 5368 *C. reinhardtii* proteins and 1737 *M. japonicum* proteins. In the unfractionated mono-mix experiment (Sample Set 2), it included 6246 *C. reinhardtii* proteins and 2128 *M. japonicum* proteins. In the fractionated mono-mix experiment (Sample Set 2, fractionated), it included 9934 *C. reinhardtii* proteins and 4179 *M. japonicum* proteins). Missing values were handled by exclusion rather than imputation. *p* values were adjusted for the number of t-tests conducted per species in each experiment using a Benjamini-Hochberg correction by applying the p.adjust function with method = "fdr". Significant differences in protein abundance were defined as those where |log$_2$-fold change| > 1, *p-adj.* < 0.05, and the mean log$_2$-transformed MASIC value was greater than the limit of quantitation in both the coculture and the monoculture control.

## Gene ontology (GO) enrichment analysis

GO term assignments for the *C. reinhardtii* v6.1 proteome were downloaded from Phytozome (https://phytozome-next.jgi.doe.gov/). GO terms were assigned *de novo* to the *M. japonicum* MAFF303099 proteome (available from the NCBI with accession GCF_000009625.1) using InterProScan (v5.59-91.0) with --iprlookup –goterms [53]. Then, the lists of proteins with significant changes in their abundance in coculture relative to monoculture were tested for GO term enrichment using the enricher function in the R package *clusterProfiler* (v3.0.4). *p* values were corrected using a Bonferroni-Holm adjustment, and enriched GO terms where *p-adj.* < 0.05 were considered significant.

## Accession numbers

Raw proteomics data have been deposited at http://massive-ftp.ucsd.edu/ under the accession MSV000098360 and are available as of the date of publication.

## Results

### Bacterial protein detection by DDA LC-MS/MS is impaired in the presence of an algal partner

To determine whether label-free DDA LC-MS/MS could be used to monitor changes to a bacterium's proteome during coculture with an alga, we grew the model alga *C. reinhardtii* and the bacterium *M. japonicum* under four conditions and performed proteomics: algal monoculture, bacterial monoculture supplemented with sucrose, coculture, and coculture supplemented with sucrose (S1 and S2 Tables). We found that the presence of *C. reinhardtii* reduced detection of bacterial proteins substantially (Fig 1). In the absence of the alga, nearly 4000 unique bacterial proteins were detected, but in coculture, the number of unique bacterial proteins detected decreased as the relative abundance of the alga increased, such that <500 bacterial proteins could be detected in the coculture in the absence of sucrose (Fig 1A and 1B). For bacterial proteins detected in all three sample types, the measured abundance was also significantly decreased proteome-wide in coculture samples relative to monoculture samples (Fig 1C, Wilcoxon rank-sum tests, $p < 0.05$). Whereas the histogram of algal protein abundances in the monoculture samples overlapped with the histograms of algal protein abundances in coculture samples (muddied color from overlapping purple, turquoise, and orange histograms), the histograms of bacterial protein abundances were shifted to lower abundances as the relative abundance of the alga increased (Fig 1C, Wilcoxon rank-sum tests, $p < 0.05$). These results suggest a systematic decrease in the selection of bacterial peptide precursor ions in the presence of relatively abundant algal peptides in coculture samples.

Similar observations were apparent upon reanalysis of proteomics data from a published study on the interaction between *C. reinhardtii* and the actinomycete *Arthrobacter* strain P2b [21]. Although bacterial density was not impacted by the presence of the alga, the number of bacterial proteins detected by label-free DDA LC-MS/MS decreased in coculture relative to monoculture (S1A Fig), and the measured abundance of bacterial proteins that were detected in both conditions was decreased globally in samples containing *C. reinhardtii* proteins (S1B Fig, Wilcoxon rank-sum tests, $p < 0.05$) [21]. We wondered if decreased detection of bacterial proteins in coculture samples was unique to those containing eukaryotic partners with larger cells, or rather generalizable to any sample with another microbe. Therefore, we also reanalyzed label-free DDA LC-MS/MS proteomics data from syntrophic bacterial cultures of *Dehalococcoides ethenogenes* strain 195 and *Desulfovibrio vulgaris* Hildenborough [54]. In this case, the two organisms are closer in both cell size and the number of protein-coding genes. In addition, coculture increased growth of *D. ethenogenes*, and *D. ethenogenes* cells made up roughly 80% of the biomass in the cocultures analyzed by proteomics [54]. Nevertheless, detection of *D. ethenogenes* proteins was markedly decreased in cocultures relative to monocultures (S1C and S1D Fig). Furthermore, a similar observation was reported from proteomic analysis of the archaeon *Nanoarchaeum equitans* in the presence and absence of another archaeon, the epibiont *Ignicoccus hospitalis* [34]. These data on diverse symbiotic interactions suggest that increased complexity of coculture samples decreases microbial protein detection regardless of the identity or relative abundance of species present.

We sought to assess how differences in detection influence differential expression analysis of algal and bacterial proteins. We used standard criteria for defining significant changes in protein abundance: $|\log_2\text{-fold change}| > 1$ and *p-adj.* $< 0.05$ from a Welch two sample t-test of the protein's abundance (MASIC values) in replicate coculture and monoculture samples. We found that no *C. reinhardtii* proteins showed significant changes in abundance in coculture relative to monoculture (S2A Fig). This was not surprising, as we previously showed that wild-type *C. reinhardtii* growth and physiology are not significantly impacted by *M. japonicum* [26]. When we compared the abundance of the 1737 *M. japonicum* proteins detected in bacterial monocultures and cocultures with sucrose, we found that 1350 proteins (78%) were significantly decreased in abundance in the coculture samples, whereas only 12 (0.7%) were increased (S2B Fig). Downregulation of such a large proportion of the bacterium's proteome was not expected, particularly since *M. japonicum's* preferred carbon source, sucrose, was provided in both conditions. Given the systematic depression of bacterial protein abundance observed in Fig 1C, it was not possible to distinguish whether these differences in protein abundance were biological or methodological.

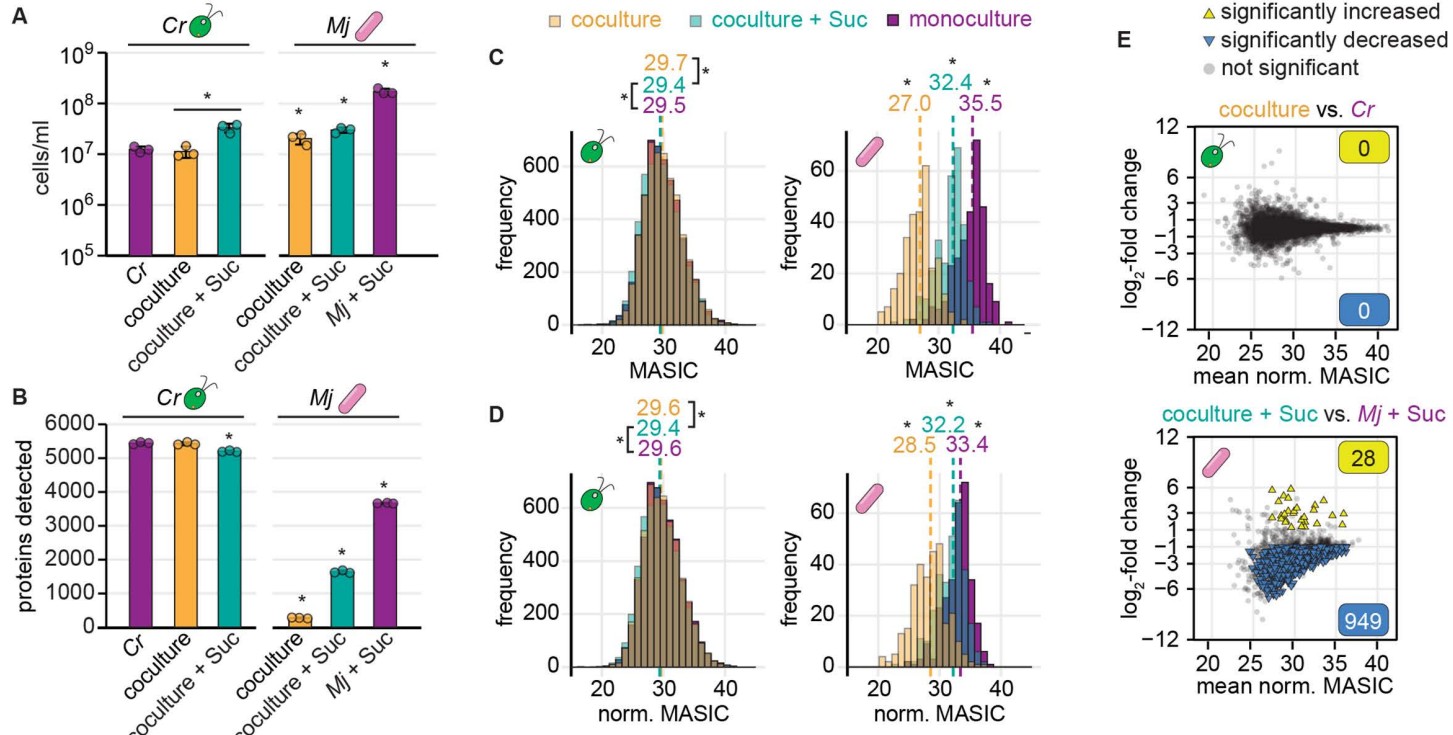

**Fig 1. Detection of bacterial proteins decreases in coculture samples.** Triplicate continuous-light-grown monocultures and cocultures with and without sucrose were collected for LC-MS/MS proteomics. See also S1 and S2 Figs. **(A)** Density of *C. reinhardtii* cells (left) and *M. japonicum* CFU (right) in triplicate monocultures (purple), cocultures (orange), and cocultures with sucrose (turquoise) upon collection for proteomics. The bars represent the mean of three replicate cultures, and error bars represent the standard deviation from the mean. Asterisks indicate significant differences by two-tailed Student's t-tests ($p < 0.05$). **(B)** Number of unique *C. reinhardtii* proteins (left) and *M. japonicum* proteins (right) detected in samples from **(A)** by LC-MS/MS proteomics. Asterisks indicate significant differences as in **(A)**. **(C)** Distribution of algal (left) and bacterial protein abundances (right) in cocultures (transparent orange), cocultures with sucrose (transparent turquoise), and monocultures (purple) for proteins detected in all three conditions. When distributions overlap, the color is muddied. The unnormalized MASIC values were averaged across the three biological replicates. Median values are shown above the dashed lines. Asterisks next to the medians indicate significant differences between the indicated distributions by a Wilcoxon rank-sum test ($p < 0.05$). **(D)** Distribution of algal (left) and bacterial protein abundances (right) in cocultures (transparent orange), cocultures with sucrose (transparent turquoise), and monocultures (purple) after quantile normalization, presented as in **(C)**. **(E)** Changes in quantile-normalized protein abundances in coculture relative to monoculture for *C. reinhardtii* (left) and *M. japonicum* (right). Significant differences (triangles) were defined as those where |log₂-fold change| > 1, *p-adj.* < 0.05 from a Welch two sample t-test of the quantile-normalized MASIC values, and the mean MASIC value was greater than the limit of quantitation in both the coculture and the monoculture.

Data normalization has been shown to be important for differential expression analysis using label-free DDA proteomics [55–58], and it has been employed in order to perform differential expression analysis on data collected from microbial cocultures and monocultures [34,36,59]. Therefore, we tested whether data normalization could allow for appropriate comparisons of the abundance of bacterial proteins in monoculture and algal coculture. We applied quantile normalization, a complete normalization strategy that equalizes sample distributions by ranking the values in each sample and substituting the values by the rank-wise means [51]. This strategy has been shown to optimally reduce technical variability in shotgun LC-MS/MS datasets [56]. Since the distributions of algal protein abundances were markedly different from the distributions of bacterial protein abundances (Fig 1C), proteomics data from each organism were quantile normalized independently: the abundances of each organism's proteins in each replicate sample were quantile normalized relative to the abundances of that organism's proteins in all relevant samples. For example, the abundances of *M. japonicum* proteins in Replicate A of the monoculture condition were quantile normalized together with the abundances of *M. japonicum*

proteins in the other monoculture replicate samples, in the replicate samples from cocultures, and in the replicate samples from cocultures with sucrose, but not with the abundances of proteins in the *C. reinhardtii* monoculture samples. We found that this per-organism quantile normalization strategy led to greater overlap in the distributions of protein abundances for both species (Fig 1D, a greater proportion of the histograms are overlapping and their color is muddied; S2B Table). However, significant differences in the distributions remained (Fig 1D, Wilcoxon rank-sum tests, $p < 0.05$). Furthermore, differential expression analysis using the quantile-normalized values still suggested that most bacterial proteins were significantly decreased in abundance in coculture relative to monoculture in the presence of sucrose (Fig 1E). In this case, 949 bacterial proteins (55%) decreased in abundance in coculture samples, whereas only 28 bacterial proteins (1.6%) increased. Thus, comparison of protein abundance determined by shotgun LC-MS/MS analysis of monoculture and algal coculture samples is not an appropriate way to discern symbiosis-induced changes in the bacterial proteome, even after data normalization.

## The "mono-mix" strategy allows comparable detection of bacterial proteins from a coculture and a monoculture control by DDA proteomics

In order to enable differential expression analysis of the bacterial proteome in cocultures relative to monocultures from label-free DDA, we sought to develop a more appropriate monoculture control that could emulate the analyte complexity of the coculture condition. In a previous study investigating the interaction between the archaeon *Methanococcus maripaludis* and the bacterium *D. vulgaris*, protein abundances in coculture were compared to abundances in a "synthetic blend," where equal amounts of total protein from each organism had been mixed together prior to analysis [60]. Then, the fold change in each protein's abundance in cocultures relative to the synthetic blend was multiplied by a constant correction factor according to the estimated relative abundance of the organisms in coculture. This correction factor approach assumes that differences in organism relative abundance would impact detection equally for all peptides regardless of their physicochemical properties or their abundances. Proteins encoded by *C. reinhardtii* and *M. japonicum* exhibit a wide range of isoelectric points, hydropathies, and molecular weights, which can influence their detection (S3 Fig).

We developed a modified experimental design that we term the "mono-mix" strategy, in which DDA peptide detection in control samples may more closely emulate detection in coculture samples. We grew *C. reinhardtii* monocultures and *M. japonicum* monocultures supplemented with sucrose in parallel to cocultures in minimal medium. Then, immediately prior to collecting cell pellets for proteomics, we combined the two monocultures to approximate the bacteria-to-algae ratio expected in the cocultures (Fig 2A). We call this combined monoculture the "mono-mix" control.

Like other algae, *C. reinhardtii* coordinates its metabolism and gene expression with the time of day under diurnal light. The alga performs photosynthesis, cell growth, and starch synthesis during the day period, initiates replication at dusk, then degrades starch and induces fermentative metabolism during the night in the G0 phase of the cell cycle [41,61]. To assess whether these diurnal rhythms influence *M. japonicum* in coculture, we applied the "mono-mix" strategy to cultures grown under continuous light and diurnal light. *C. reinhardtii* strain CC-5390 was used for this experiment, as this strain is amenable to synchronization under diurnal cycles. This yields populations of cells exhibiting nearly the same physiological state and gene expression, which provide exceptional signal-to-noise for resolving diurnal patterns [41]. We sampled cultures grown in diurnal light at both the end of the night (36 h after inoculation) and the end of the day (48 h after inoculation), and we compared them to cultures grown in continuous light sampled 36 h after inoculation (Fig 2A, S1 and S3 Tables).

Preparing a mono-mix control with the same bacteria-to-algae ratio as the parallel coculture was challenging. We determine the number of bacterial cells in a culture as CFU on TY agar plates, which is assessed only after 4 d of incubation. Thus, only a rough estimate of the bacterial cell density is available at the time samples are collected for proteomics. Therefore, there was variation in how comparable the cell densities were in the mono-mix controls and their parallel cocultures (Fig 2B). We found that although the number of bacterial proteins detected in the cocultures (roughly 800–2200)

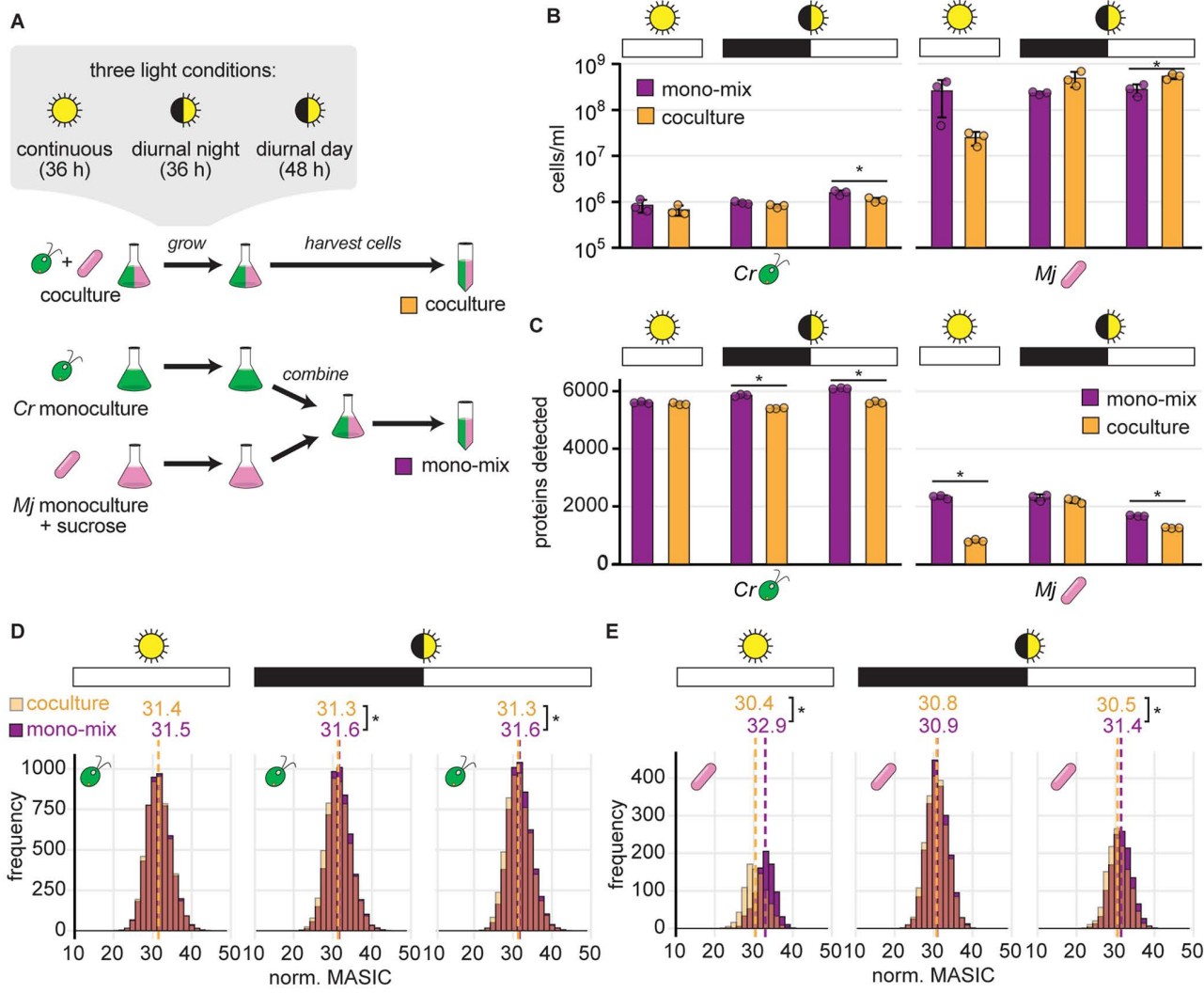

**Fig 2. The mono-mix strategy enables comparable DDA detection of bacterial proteins in the coculture and the monoculture control when a similar bacteria-to-algae ratio is achieved.** To assess whether *C. reinhardtii's* diurnal rhythms influence *M. japonicum*, cultures were grown in either continuous or diurnal light. See also S3 and S4 Figs. **(A)** Schematic of experiment. Triplicate cocultures of *C. reinhardtii* (green) and *M. japonicum* (pink) were grown in parallel with triplicate monocultures of *C. reinhardtii* and *M. japonicum* with 150 µg/ml sucrose. Then, the *M. japonicum* monocultures were added to the *C. reinhardtii* monocultures to achieve a "mono-mix" control (purple) with a similar bacteria-to-algae ratio as the coculture (orange). Cultures were grown in either continuous (sun icon) or diurnal light (eclipsed sun icon). Continuous light cultures were collected 36 h after inoculation, and the diurnal light cultures were collected at the end of the night (36 h after inoculation) and the end of the day (48 h after inoculation). **(B)** Density of *C. reinhardtii* cells and *M. japonicum* CFU in triplicate cocultures (orange) and mono-mix controls (purple) grown in continuous light (sun icon) or diurnal light (eclipsed sun icon) upon collection for proteomics at the end of the dark or light phases (black or white bars, respectively). The bars represent the mean of three replicate cultures, and error bars represent the standard deviation from the mean. Asterisks indicate significant differences by two-tailed Student's t-tests ($p < 0.05$). **(C)** Number of unique algal (left) and bacterial proteins (right) detected in the samples from **(B)**. Asterisks indicate significant differences as in **(B)**. **(D)** Distribution of normalized abundances of algal proteins in cocultures (transparent orange) and mono-mix controls (purple) grown in continuous light (sun icon) or diurnal light (eclipsed sun icon) collected at the end of the dark or light phases (black or white bars, respectively). When distributions overlap, the color is muddied. The quantile-normalized MASIC values were averaged across the three biological replicates. Median values are shown above the dashed lines. Asterisks next to the medians indicate significant differences between the indicated distributions by a Wilcoxon rank-sum test ($p < 0.05$). **(E)** Distribution of normalized abundances of bacterial proteins presented as in **(D)**.

was more comparable to that of the mono-mix controls (roughly 1600–2300), it was still significantly lower in the samples collected from the continuous light cultures and the diurnal light cultures at the end of the day (Fig 2C, Student's t-test, $p<0.05$). The mono-mix control was most comparable to the coculture in the diurnal night samples, both in terms of organism relative abundance and of the number of bacterial proteins observed (Fig 2B and 2C). In this condition, there was no significant difference in the total number of bacterial proteins detected (Fig 2C, Student's t-test, $p=0.48$) nor in the distribution of bacterial protein abundances in the coculture relative to the mono-mix control (Fig 2E, Wilcoxon rank-sum test, $p=0.30$). This was true whether data were quantile normalized or not (Fig 2E and S4B Fig, respectively).

## Data normalization and comparison to mono-mix controls reveal that light regime may influence the *M. japonicum* proteome during coculture with *C. reinhardtii*

Having achieved more comparable detection of the bacterial proteome in cocultures and mono-mix controls, we next sought to determine whether these data could reveal the impact of light regime and time of day on the organisms during their interaction. We performed *k*-means clustering and principal component analysis (PCA) on the quantile-normalized abundances of proteins that were detected in all conditions (the cocultures and mono-mix controls in continuous light, diurnal night, and diurnal day). The *C. reinhardtii* proteome could be separated into 6 clusters that exhibited distinct patterns with respect to light regime and the presence of the bacterium (Fig 3A, S3B Table). For example, Cluster Cr1 proteins were more abundant in coculture under all light regimes, and Cluster Cr5 proteins were more abundant in samples collected during the light phase (either continuous or diurnal) regardless of the presence of the bacterium during growth.

On the other hand, the *M. japonicum* proteome could only be separated into 2-to-3 distinct clusters (Fig 3B, S3B Table). There was only a subtle difference in the pattern of expression between bacterial proteins in Clusters Mj1 and Mj2 across the different light and culture conditions. Most bacterial proteins appeared to have lower abundance in the continuous light coculture samples (for which the distribution of protein abundances was significantly lower, Fig 2E, Wilcoxon rank-sum test, $p<2\text{x}10^{-16}$) and high abundance in the continuous light mono-mix controls and the diurnal light cultures collected at the end of the night. Bacterial proteins in Cluster Mj3 appeared higher in cocultures than in monocultures, particularly in the diurnal night samples. These results suggest that complexity-related differences in detection of bacterial peptides in the various samples still influence the data.

PCA revealed that 27% of the variation in *C. reinhardtii's* proteome could be explained by the presence or absence of light, and 13% of the variation was driven by the interaction with the bacterium (Fig 3C). PCA of *M. japonicum's* proteome showed that the continuous light cocultures (white triangles) were most distinct from their respective mono-mix controls (white circles) (Fig 3D). This was also evident in the *k*-means clustered heatmap, where the majority of bacterial proteins detected appeared less abundant in the cocultures grown in continuous light than in the mono-mix controls grown in continuous light (Fig 3B). As discussed, bacterial protein detection was significantly lower in the continuous light cocultures relative to their mono-mix controls (Fig 2C and 2E). In addition, differential expression analysis suggested that in continuous light, many more proteins were significantly decreased in abundance in coculture than were increased (S5 Fig, S4 Table). Therefore, the patterns observed by clustering analyses likely reflect differences in organism relative abundance and consequent differences in peptide detection in the coculture and mono-mix samples collected from continuous light.

Interestingly, however, PCA of *M. japonicum's* proteome also revealed that while the mono-mix controls (circles) clustered relatively close to one another on both the first and second principal components, the cocultures (triangles) were quite separated from one another by the first two principal components (Fig 3D). This suggested that light regime had a greater influence on the bacterial proteome in coculture than it did in monoculture.

Comparative analysis of the significant changes in bacterial protein abundance revealed that the majority of changes occurred uniquely in either continuous light, diurnal night, or diurnal day (Fig 3E, S5 Fig, and S4 Table). Two bacterial proteins were significantly increased regardless of light regime: WP_010914712.1, annotated as an extracellular solute-binding protein, and WP_010912842.1, annotated as a sugar-binding protein. Light-insensitive changes in proteins with

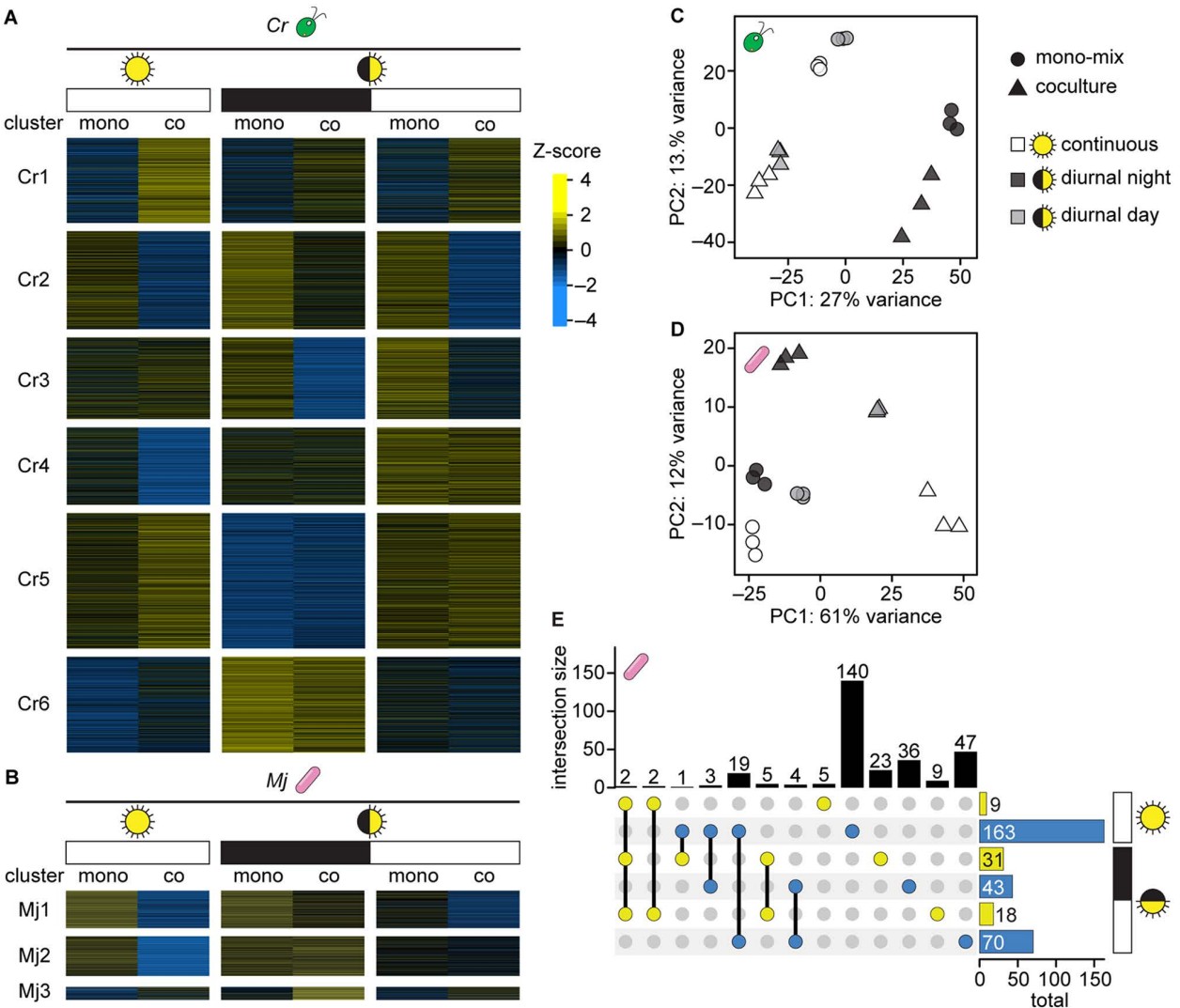

**Fig 3. Data normalization and comparison to mono-mix controls reveals that light regime may influence the *M. japonicum* proteome during coculture with *C. reinhardtii*. (A)** *k*-means clustering (*k*=6) of the Z-score of quantile-normalized mean algal protein abundances in the mono-mix experiment. 4737 *C. reinhardtii* proteins that were detected in at least two biological replicates in all conditions were included in the analysis. **(B)** *k*-means clustering (*k*=3) of the Z-score of quantile-normalized mean bacterial protein abundances in the mono-mix experiment. 828 *M. japonicum* proteins that were detected in at least two biological replicates in all conditions were included in the analysis. **(C)** PCA of quantile-normalized protein abundances for the 3817 *C. reinhardtii* proteins that were detected in all samples: three biological replicates of the mono-mix (circles) and coculture samples (triangles) under continuous light (white), diurnal night (dark grey), and diurnal day (light grey). **(D)** PCA of quantile-normalized protein abundances for the 569 *M. japonicum* proteins that were detected in all samples, presented as in **(C)**. **(E)** Intersections and differences in the bacterial proteins that were significantly increased (yellow) and significantly decreased (blue) in samples collected from continuous light, diurnal night, and diurnal day. Intersection sizes (indicated by lines connecting colored dots) and differences (indicated by single colored dots) are shown as the vertical bars, while the total number of bacterial proteins that are significantly increased and decreased are shown as the horizontal bars. Significant differences were defined as those where |log$_2$-fold change|>1, *p-adj.*<0.05 from a Welch two sample t-test of the quantile-normalized MASIC values, and the mean MASIC value was greater than the limit of quantitation in both the coculture and the monoculture. See also S5 Fig.

such functions could be expected in coculture, where the bacterium relies on *C. reinhardtii* for reduced carbon substrates, compared to monocultures grown with sucrose, *M. japonicum's* preferred carbon source.

Taken together, these results suggest that although complexity-related differences in peptide detection still influence proteomic comparisons between cocultures and monocultures, the mono-mix strategy together with data normalization

may allow one to capture more meaningful changes in the proteome resulting from an algal-bacterial interaction by label-free DDA.

## Fractionation improves detection and differential expression analysis of sparse proteins in coculture

Sample fractionation is a strategy that physically separates analytes in complex samples (e.g., by liquid chromatography (LC)) to reduce the diversity of ions entering the mass spectrometer at any given moment [62–66]. This simplification enables the instrument to perform additional fragmentation scans to identify and quantify detectable compounds, decreasing the impact of complexity, particularly for low-abundance analytes.

To test whether fractionation could improve detection of bacterial proteins from coculture by label-free DDA, we analyzed the proteome of a coculture sample after separation into 1, 6, 12, or 24 fractions. Each level of fractionation substantially increased the number of proteins that could be detected in coculture, both for *M. japonicum* as well as for *C. reinhardtii* (Fig 4A). We next sought to combine fractionation with the mono-mix strategy to determine changes in the *M. japonicum* proteome upon interaction with the alga. Since the diurnal night samples had the most comparable cocultures and mono-mix controls (Fig 2), we chose these samples as our test-case and analyzed them each as a single fraction or as 12 fractions by label-free DDA (S1 and S5 Tables). This level of fractionation resulted in a 1.8-fold increase in the number of algal proteins detected and a 1.9-fold increase in bacterial proteins detected (Fig 4B). In addition, the fractionated samples showed no significant difference in the distribution of protein abundances between the coculture and the respective mono-mix controls for either organism (Fig 4C bottom, Wilcoxon rank-sum tests, $p > 0.05$).

We next determined how combining fractionation with the mono-mix strategy influenced differential expression analysis of cocultures relative to monocultures. We found that running samples as 12 fractions nearly doubled the number of proteins that met the detection criteria for differential expression analysis. Over 4000 more algal proteins and 2000 more bacterial proteins could be assessed for changes in coculture relative to mono-mix controls when samples were fractionated (Fig 5A and 5B). Most of these newly captured proteins were of lower abundance on average.

While sample fractionation did not reveal any additional significant changes in the algal proteome (Fig 5A), nearly 4X the number of significant changes in bacterial protein abundance could be observed if samples were fractionated (Fig 5B and S6 Fig, S6 Table). Of the 55 *M. japonicum* proteins that exhibited a significant change in abundance in coculture relative to the mono-mix controls, 15 did not meet our detection criteria without sample fractionation. The remaining 40 proteins were detected in samples regardless of fractionation, but a significant change ($|\log_2$-fold change$| > 1$, *p-adj.* $< 0.05$) was only observable if the samples were fractionated. Thus, sample fractionation improves differential expression analysis of bacterial proteins in coculture not only by increasing the detection of lowly abundant proteins, but also by increasing the signal-to-noise, such that changes in abundance can be observed.

Of the 36 significant increases in protein abundance in coculture observed if samples were analyzed as 12 fractions, 34 of them were missed if samples were analyzed as 1 fraction (Fig 5C). Gene ontology (GO) enrichment analysis revealed that in addition to "outer membrane-bounded periplasmic space" proteins, *M. japonicum* increased the abundance of groups of proteins significantly enriched for transport, thiamine-binding, and carbohydrate-binding activity (Fig 5D). These additional changes detectable upon sample fractionation may provide clues about the kinds of metabolites that are exchanged between *C. reinhardtii* and the bacterium.

## Discussion

Here, we have documented a concerning artifact in label-free DDA LC-MS/MS data from mixed algal-bacterial cocultures, developed several strategies to minimize and even overcome this artifact, and enabled more meaningful differential expression analysis of algal and bacterial proteins in cocultures relative to monocultures from such data (Fig 6). We found that detection of bacterial proteins was decreased in the presence of an alga, both in terms of the number of bacterial proteins detected and in terms of their measured abundances (Fig 1). Standard differential expression analysis on these data

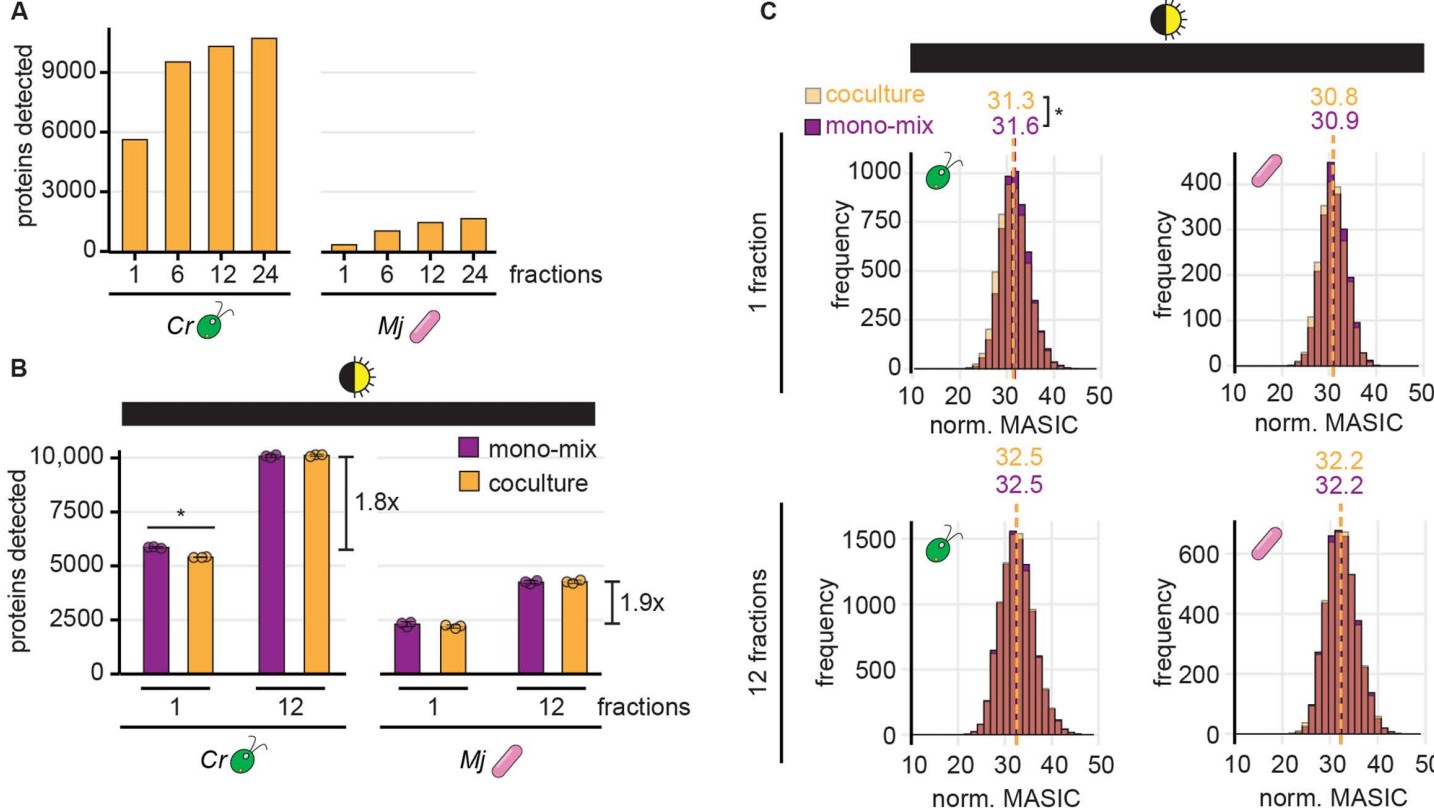

**Fig 4. Fractionation improves detection of sparse bacterial proteins in coculture. (A)** The number of unique algal (left) and bacterial proteins (right) detected when a coculture sample was analyzed as 1, 6, 12, or 24 fractions. **(B)** The number of unique algal (left) and bacterial proteins (right) detected from cocultures (orange) and mono-mix controls (purple) collected at the end of the night from diurnal light-grown cultures when analyzed as 1 or 12 fractions. Bars represent the mean of three replicate cultures, and error bars represent the standard deviation from the mean. Asterisks indicate significant differences by two-tailed Student's t-tests ($p < 0.05$). **(C)** Distribution of normalized abundances of algal (left) and bacterial proteins (right) in cocultures (transparent orange) and mono-mix controls (purple) collected at the end of the night when analyzed as 1 (top) or 12 fractions (bottom). When distributions overlap, the color is muddied. The quantile-normalized MASIC values were averaged across the three biological replicates. Median values are shown above the dashed lines. Asterisks next to the medians indicate significant differences between the indicated distributions by a Wilcoxon rank-sum test ($p < 0.05$).

comparing monocultures and cocultures with sucrose suggests that *M. japonicum* downregulates 78% of its proteome in the presence of *C. reinhardtii* – an artifact that reflects increased suppression of bacterium-derived peptide ions by abundant alga-derived peptide ions in mixed samples (S2 Fig). Reanalysis of publicly available label-free DDA proteomics data from other alga-bacterium and bacterium-bacterium interactions suggests that increased complexity in coculture samples decreases microbial protein detection regardless of the identity or relative abundance of species present (S1 Fig) [21,54].

We first assessed an *in silico* data normalization strategy to improve DDA proteomic analysis of the algal-bacterial interaction, as has been suggested in the literature [36]. Quantile normalization per organism led to greater overlap in the distributions of protein abundances in coculture samples and monoculture samples (Fig 1C–1D). However, differential expression analysis using quantile-normalized values still suggested that most bacterial proteins were significantly decreased in abundance in coculture relative to monoculture (Fig 1E). Thus, we found that data normalization alone does not eliminate the influence of altered detection on differential expression analysis of sample types with different relative abundances of two organisms.

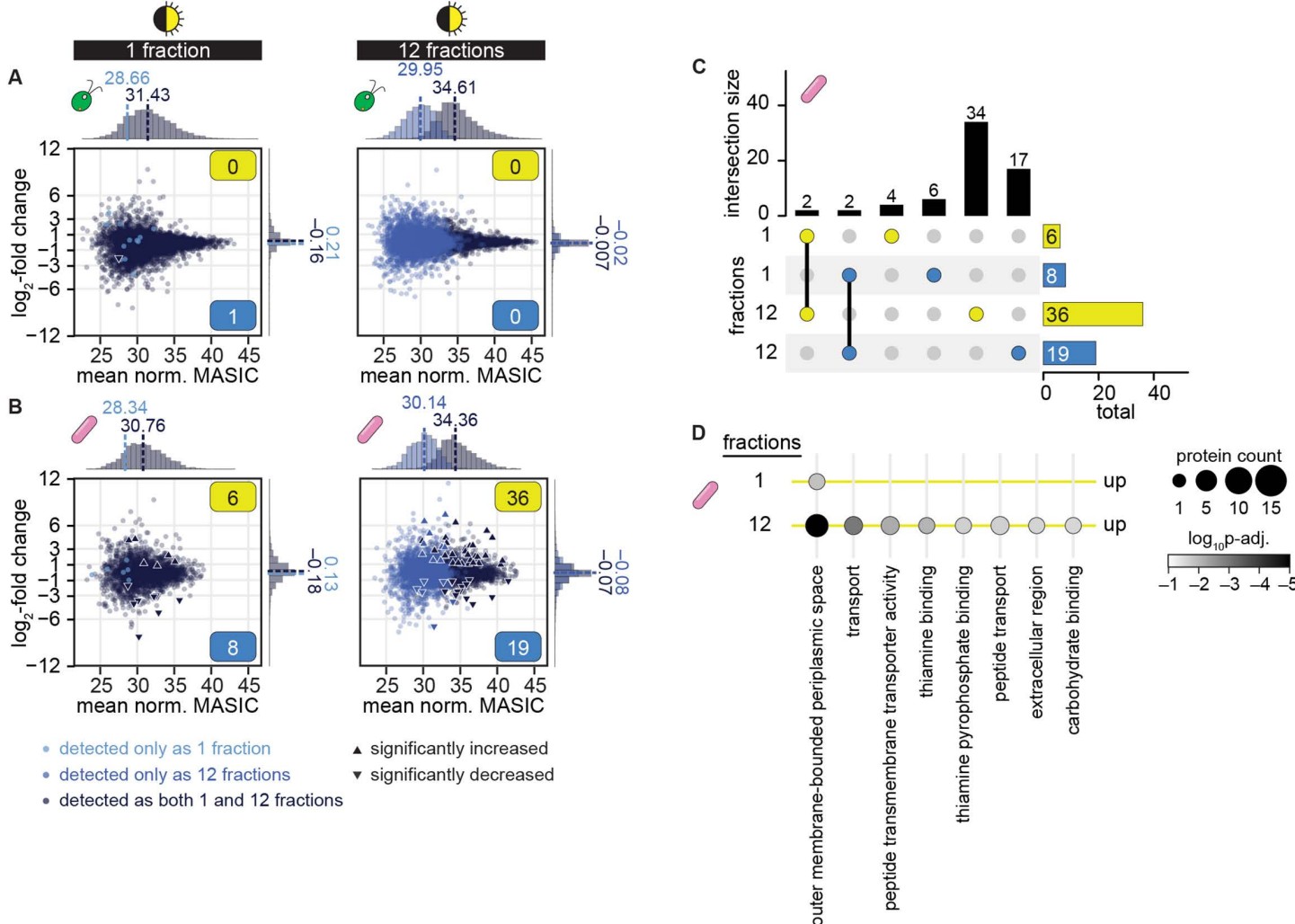

**Fig 5. Sample fractionation improves differential expression analysis of proteins in cocultures relative to mono-mix controls.** See also S6 Fig.
**(A)** Changes in quantile-normalized *C. reinhardtii* protein abundance in coculture relative to mono-mix controls at the end of the night when samples were analyzed as 1 (left) or 12 fractions (right). Points are colored based on their detection criteria: lighter blue indicates that the protein only met the criteria when the sample was analyzed as either 1 or 12 fractions, navy blue indicates that the protein met the criteria when the sample was analyzed both as 1 and as 12 fractions. Significant differences (triangles) were defined as those where |$\log_2$-fold change| > 1, *p-adj.* < 0.05 from a Welch two sample t-test of the quantile-normalized MASIC values, and the mean MASIC value was greater than the limit of quantitation in both the coculture and the mono-mix. Yellow and blue boxes indicate the number of significantly increased and decreased proteins, respectively. Histograms show the distribution of mean protein abundances (*x* axes) and of $\log_2$-fold changes (*y* axes) for proteins that only met the detection criteria when the sample was analyzed as either 1 or 12 fractions (lighter blue) and for proteins that met the detection criteria when the sample was analyzed both as 1 and as 12 fractions (navy blue); median values are shown above the dashed lines. **(B)** Changes in quantile-normalized *M. japonicum* protein abundance in coculture relative to mono-mix controls at the end of the night when samples were analyzed as 1 (left) or 12 fractions (right) presented as in **(A)**. **(C)** Intersections and differences in the bacterial proteins that were significantly increased (yellow) and significantly decreased (blue) in coculture when samples were analyzed as 1 or 12 fractions. Intersection sizes (indicated by lines connecting colored dots) and differences (indicated by single colored dots) are shown as the vertical bars, while the total number of bacterial proteins that are significantly increased and decreased are shown as the horizontal bars. **(D)** GO term enrichment in bacterial proteins that were significantly increased in quantile-normalized abundance in coculture when samples were analyzed as 1 or 12 fractions. The number of proteins with the corresponding GO term in each comparison is indicated by dot size and the *p-adj.* by the shading. No GO terms were significantly enriched in the bacterial proteins that were significantly decreased in quantile-normalized abundance in coculture, regardless of sample fractionation.

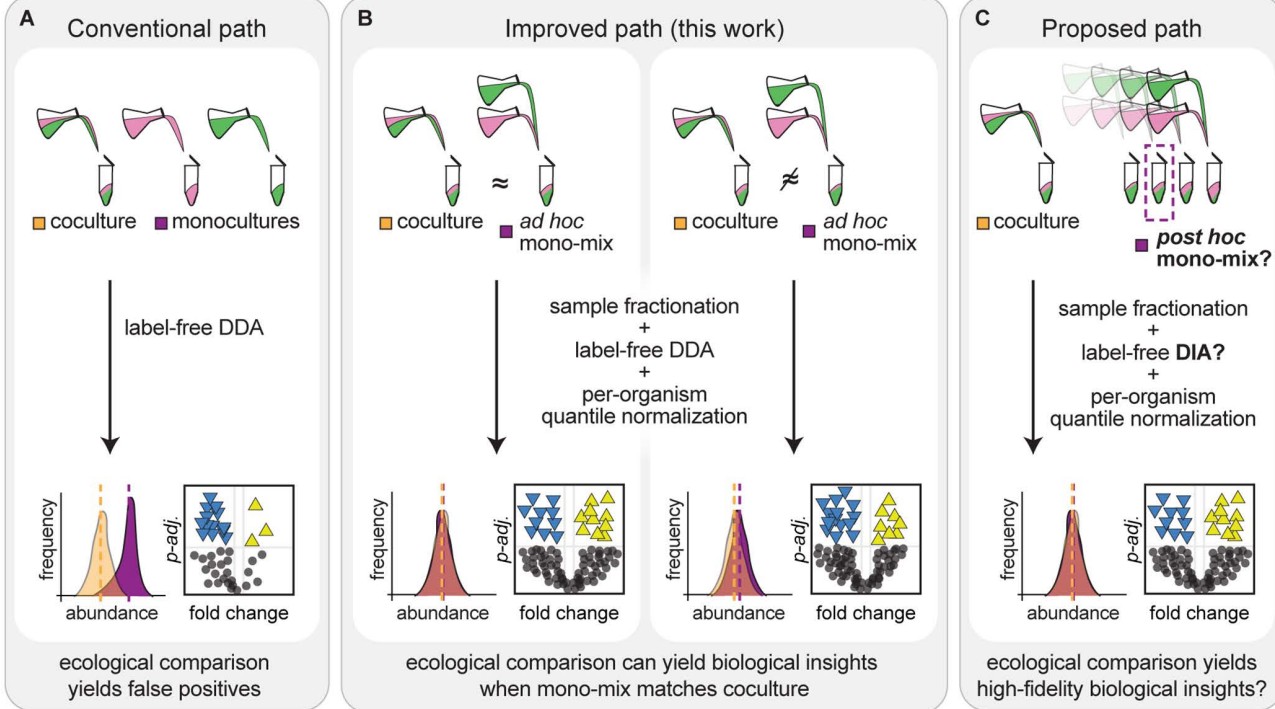

**Fig 6. Graphic representation of challenges in comparative proteomics of microbial symbioses and strategies for overcoming them. (A)** During conventional label-free DDA LC-MS/MS proteomics, increased complexity of coculture samples decreases protein detection in coculture (orange histogram, dotted line represents the median) relative to monoculture (purple histogram, dotted line represents the median). As a result, differential expression analysis of proteins in coculture relative to monoculture results in false-positive proteins with significantly lower abundance in coculture (blue triangles in volcano plot). **(B)** In the modified experimental design developed in this work, coculture samples are compared to a "mono-mix" control where upon harvest, monocultures are combined to the predicted relative abundance of the cocultures *ad hoc*. In the improved analytical framework, label-free DDA coverage is increased through peptide sample fractionation, and protein abundance data are quantile normalized per organism. Comparative analysis of the ecological conditions yields biological insights when the relative abundance of the mono-mix matches that of the coculture (left panel), but false positives arise if it does not (right panel). **(C)** For future work, we propose preparation of several mono-mix samples of various target relative abundances upon harvest and *post hoc* selection of the mono-mix whose relative abundance best matches the parallel coculture (dotted purple box). We also propose data-independent acquisition (DIA) LC-MS/MS proteomics to allow unbiased, high coverage of the proteomes of multiple microbes from complex mixed culture samples.

To enable differential expression analysis of DDA data from mixed algal-bacterial samples, we developed a modified experimental design that we term the "mono-mix" strategy, where *C. reinhardtii* monocultures and *M. japonicum* monocultures are combined into a single mono-mix control that emulates the bacteria-to-algae ratio of the coculture sample to which they will be compared (Fig 6B). Using this strategy, detection of bacterial proteins was more comparable in cocultures and monoculture controls (Fig 2). However, we found that even small differences in relative abundance still appear to shape peptide detection and differential expression analysis.

It was challenging to achieve comparable bacteria-to-algae ratios in the *ad hoc* mono-mix samples, as bacterial density is estimated *post hoc* from CFU counts of viable cells. Furthermore, previous work has shown that these counts are not an accurate reflection of bacterial biomass in this system, as *M. japonicum* undergoes reductive division when in coculture with *C. reinhardtii* [26]. The bacterium continues to divide despite reduced biomass synthesis, resulting in smaller cells. For future work, we recommend preparing several mono-mix samples of various target relative abundances and *post hoc* selecting the samples whose actual relative abundance of biomass best matches the parallel coculture for comparison (Fig 6C). Alternatively, monoculture samples could be frozen and then combined based on measurements of relative abundance

and cellular protein concentration just prior to sample processing. In addition, fluorescence in situ hybridization (FISH), flow cytometry, or total organic carbon measurements may be used to more accurately estimate bacterial biomass [67].

Even though complexity-related differences in peptide detection still influenced the data, the mono-mix experiment revealed that light regime shapes the *M. japonicum* proteome during coculture with *C. reinhardtii*, but not during monoculture (Fig 3). *C. reinhardtii* exhibits massive changes in its proteome and metabolome over diurnal cycles [41,68,69]. Previous work showed that the exchange of reduced carbon between *C. reinhardtii* and *M. japonicum*, as well as the resulting amount of *M. japonicum* growth, is reduced in diurnal light compared to continuous light [26]. Unsurprisingly, we found that the bacterial proteome was distinct in continuous light and diurnal light, likely reflecting more drastic carbon limitation in the latter situation (Fig 3D). Interestingly though, we found that the bacterial proteome is also quite distinct in the day and night phases of the same diurnal period, showing that *M. japonicum's* proteome is shaped by *C. reinhardtii's* diurnal program. Future work can further disentangle the biological responses from detection-related differences.

Finally, we augmented the mono-mix approach by using sample fractionation to improve detection of lowly abundant bacterial proteins. By analyzing peptide samples divided into 12 fractions, we achieved over a 1.8-fold increase in the number of proteins that could be detected from either organism (Fig 4B). Even analyzing samples as 6 fractions allows a substantial increase in protein detection (Fig 4A), and thus users may weigh the costs and benefits of fractionation level depending on the number of experimental conditions. Not only did sample fractionation increase the number of proteins detected, it also increased the signal-to-noise of measurements, capturing nearly 4X the number of significant changes in bacterial protein abundances (Fig 5). This included an increase in proteins with peptide transport, carbohydrate binding, and thiamine binding activities. Thus, our proteomic analysis has yielded novel clues about an algal-bacterial interaction.

Our analysis recognizes and emphasizes the potential for misinterpreting proteomic comparisons of systems that differ in their ecological complexity. The problem is not unique to this study or these organisms, since the phenomenon is evident also in independent work in the literature [21,34,36,54,60]. Furthermore, we found that even subtle differences in the relative abundance of an alga and a bacterium across different samples could lead to differences in peptide detection (Fig 2). Thus, we predict that in addition to monocultures and cocultures, any two treatments that exhibit substantial differences in the relative abundance of interacting organisms would suffer from altered detection by DDA that could result in false positives during differential expression analysis.

Here, we have shown that when used in tandem, data normalization, sample fractionation, and the mono-mix strategy enable discrimination of meaningful changes that arise during a microbial symbiosis from label-free DDA proteomics. These simple strategies may also strengthen other approaches that provide deeper coverage of the proteome. For example, an accurate mass and time (AMT) tag approach could be employed [70,71], in which an initial shotgun LC-MS/MS analysis with the high mass accuracy of Fourier transform ion cyclotron resonance (FTICR) MS could be performed on monoculture samples to generate a database of peptide markers for each species of interest. AMT databases are used to identify peptides by their characteristic accurate mass and elution times during LC-MS alone, allowing detection of many more peptides [70,71]. Alternatively, data-independent acquisition (DIA) LC-MS/MS proteomics could be used, in which powerful instrumentation enables fragmentation and monitoring of all precursor ions within a given mass scan [72,73]. DIA has recently been used to improve proteomic coverage of complex microbial community samples [72–76]. We posit that such approaches applied in conjunction with the mono-mix strategy would not only allow unbiased, high coverage of the proteomes of multiple microbes from mixed culture samples, but also high-fidelity determination of significant changes in protein abundance resulting from symbiosis.

## Supporting information

**S1 Fig. Bacterial protein detection is decreased in other coculture proteomic studies; related to** Fig 1 . (A) Number of unique *Arthrobacter* proteins with at least two peptides detected in at least one biological replicate of monocultures (purple) and cocultures with *C. reinhardtii* (orange) 19 h and 168 h after inoculation by LC-MS/MS proteomics reported in Windler *et al.* 2022 [21]. Spectra were mapped to the *C. reinhardtii* v5.6 genome annotation (17,741 protein-coding genes)

and *Arthrobacter* strain P2b genome annotation (4,475 protein-coding genes). (B) Distribution of *Arthrobacter* protein abundances in monocultures (purple) and cocultures with *C. reinhardtii* (transparent orange) for proteins detected in both conditions reported in Windler *et al.* 2022 [21]. When distributions overlap, the color is muddied. Median values are shown above the dashed lines. Asterisks next to the medians indicate significant differences between the indicated distributions by a Wilcoxon rank-sum test ($p < 0.05$). (C) Number of unique *Dehalococcoides ethenogenes* proteins detected in at least two biological replicates of monocultures (purple) and cocultures with *Desulfovibrio vulgaris* (orange) by LC-MS/MS proteomics reported in Men *et al.* 2012 [54]. Spectra were mapped to a concatenated database of amino acid sequences from *D. ethenogenes* strain 195 (~1,500 protein-coding genes) and *D. vulgaris* strain Hildenborough (~3200 protein-coding genes). (D) Distribution of *D. ethenogenes* protein abundances in monocultures (purple) and cocultures with *D. vulgaris* (transparent orange) for proteins detected in both conditions reported in Men *et al.* 2012 [54], presented as in (B). (TIF)

**S2 Fig. Global decrease in bacterial protein detection complicates differential expression analysis, especially prior to data normalization; related to** Fig 1 . Triplicate continuous-light-grown monocultures and cocultures with and without sucrose were collected for LC-MS/MS proteomics. (A) Changes in unnormalized algal protein abundances in coculture relative to monoculture. Significant differences (triangles) were defined as those where |$\log_2$-fold change| > 1, *p-adj.* < 0.05 from a Welch two sample t-test of the unnormalized MASIC values, and the mean MASIC value was greater than the limit of quantitation in both the coculture and the monoculture. (B) Changes in unnormalized bacterial protein abundances presented as in (A). (C) Changes in unnormalized algal protein abundance in coculture relative to mono-culture shown as a volcano plot. Significant differences (triangles) were defined as those where |$\log_2$-fold change| > 1, *p-adj.* < 0.05 from a Welch two sample t-test of the unnormalized MASIC values, and the mean MASIC value was greater than the limit of quantitation in both the coculture and the monoculture. Yellow and blue boxes indicate the number of significantly increased and decreased proteins, respectively. (D) Changes in unnormalized bacterial protein abundance in coculture relative to monoculture shown as a volcano plot as in (C). (E) Changes in quantile-normalized algal protein abundance in coculture relative to monoculture shown as a volcano plot as in (C). Significant differences (triangles) were defined as those where |$\log_2$-fold change| > 1, *p-adj.* < 0.05 from a Welch two sample t-test of the quantile-normalized MASIC values, and the mean MASIC value was greater than the limit of quantitation in both the coculture and the monoculture. Yellow and blue boxes indicate the number of significantly increased and decreased proteins, respectively. (F) Changes in quantile-normalized bacterial protein abundance in coculture relative to monoculture shown as a volcano plot as in (E). (TIF)

**S3 Fig. Physicochemical characteristics of proteins encoded by *C. reinhardtii* and *M. japonicum* vary considerably; related to** Fig 2**.** The physicochemical characteristics of *C. reinhardtii* and *M. japonicum* proteins were assessed *in silico* according to Shi *et al.* 2024 [36]. (A) Distribution of isoelectric points (pI) of proteins encoded by *C. reinhardtii* (green) and *M. japonicum* (pink). Median values are represented by the solid horizontal lines and listed above the violins, and quartiles are represented by dashed lines. (B) Distribution of hydropathy (grand average of hydropathy, GRAVY) of proteins encoded by *C. reinhardtii* (green) and *M. japonicum* (pink), presented as in (A). (C) Distribution of molecular weights of proteins encoded by *C. reinhardtii* (green) and *M. japonicum* (pink), presented as in (A). (TIF)

**S4 Fig. Distribution of unnormalized protein abundances in the mono-mix experiment; related to** Fig 2**.** Triplicate cocultures of *C. reinhardtii* and *M. japonicum* were grown in parallel with triplicate monocultures of *C. reinhardtii* and *M. japonicum* with 150 µg/ml sucrose. Then, the *M. japonicum* monocultures were added to the *C. reinhardtii* monocultures to achieve a "mono-mix" control with a similar bacteria-to-algae ratio as the coculture. Cultures were grown in either continuous or diurnal light. Continuous light cultures were collected 36 h after inoculation, and the diurnal light cultures were

collected at the end of the night (36 h after inoculation) and the end of the day (48 h after inoculation). (A) Distribution of unnormalized abundances of algal proteins in cocultures (transparent orange) and mono-mix controls (purple) grown in continuous light (sun icon) or diurnal light (eclipsed sun icon) collected at the end of the dark or light phases (black or white bars, respectively). When distributions overlap, the color is muddied. The MASIC values were averaged across the three biological replicates. Median values are shown above the dashed lines. Asterisks next to the medians indicate significant differences between the indicated distributions by a Wilcoxon rank-sum test ($p < 0.05$). (B) Distribution of unnormalized abundances of bacterial proteins presented as in (A).
(TIF)

**S5 Fig. Differential expression analysis using the mono-mix strategy may be sensitive to differences in organism relative abundance, even after normalization; related to** Figs 2 **and** 3**.** (A) Changes in quantile-normalized algal protein abundances in coculture relative to mono-mix controls when grown in continuous light (sun icon) or diurnal light (eclipsed sun icon) and collected at the end of the dark or light phases (black or white bars, respectively). Significant differences (triangles) were defined as those where |$\log_2$-fold change| > 1, *p-adj.* < 0.05 from a Welch two sample t-test of the quantile-normalized MASIC values, and the mean MASIC value was greater than the limit of quantitation in both the coculture and the monoculture. (B) Changes in quantile-normalized bacterial protein abundances in coculture relative to mono-mix controls presented as in (A). (C) Changes in quantile-normalized algal protein abundances in coculture relative to mono-mix controls when grown in continuous light (sun icon) or diurnal light (eclipsed sun icon) and collected at the end of the dark or light phases (black or white bars, respectively) shown as a volcano plot. Significant differences (triangles) were defined as those where |$\log_2$-fold change| > 1, *p-adj.* < 0.05 from a Welch two sample t-test of the quantile-normalized MASIC values, and the mean MASIC value was greater than the limit of quantitation in both the coculture and the mono-mix controls. (D) Changes in quantile-normalized bacterial protein abundances in coculture relative to mono-mix controls shown as a volcano plot as in (C).
(TIF)

**S6 Fig. Sample fractionation increases the signal-to-noise for proteomic differential expression analysis of proteins from coculture relative to mono-mix controls; related to** Fig 5**.** (A) Changes in quantile-normalized *C. reinhardtii* protein abundance in coculture relative to mono-mix controls at the end of the night when samples were analyzed as 1 (left) or 12 fractions (right). Points are colored based on their detection criteria: lighter blue indicates that the protein only met the criteria when the sample was analyzed as either 1 or 12 fractions, navy blue indicates that the protein met the criteria when the sample was analyzed both as 1 and as 12 fractions. Significant differences (triangles) were defined as those where |$\log_2$-fold change| > 1, *p-adj.* < 0.05 from a Welch two sample t-test of the quantile-normalized MASIC values, and the mean MASIC value was greater than the limit of quantitation in both the coculture and the mono-mix. Yellow and blue boxes indicate the number of significantly increased and decreased proteins, respectively. Histograms show the distribution of $\log_2$-fold changes (*x* axes) and of $\log_{10}$ *p-adj.* (*y* axes) for proteins that only met the detection criteria when the sample was analyzed as either 1 or 12 fractions (lighter blue) and for proteins that met the detection criteria when the sample was analyzed both as 1 and as 12 fractions (navy blue); median values are shown above the dashed lines. (B) Changes in quantile-normalized *M. japonicum* protein abundance in coculture relative to mono-mix controls at the end of the night when samples were analyzed as 1 (left) or 12 fractions (right) presented as in (A).
(TIF)

**S1 Table. Sample metadata for the proteomics samples; related to** S2**,** S3 **and** S5 Tables**.**
(XLSX)

**S2 Table. Algal and bacterial protein abundances under various interaction regimes; related to** Fig 1 **and** S2 Fig**.**
(XLSX)

**S3 Table. Algal and bacterial protein abundances under various interaction and light regimes determined using the mono-mix strategy; related to Figs 2–3 and S4–S5 Figs.**
(XLSX)

**S4 Table. Significant differences in protein abundance in coculture relative to mono-mix controls under various light regimes; related to Fig 3 and S5 Fig.**
(XLSX)

**S5 Table. Algal and bacterial protein abundances under various interaction regimes at the end of the night determined using the mono-mix strategy with and without sample fractionation; related to Figs 4 and 5.**
(XLSX)

**S6 Table. Significant differences in protein abundance in coculture relative to mono-mix controls at the end of the night, as determined with or without sample fractionation; related to Fig 5 and S6 Fig.**
(XLSX)

## Acknowledgments

We thank Dr. Rhona Stuart for providing data on a different symbiotic system and her encouragement of our work. We are grateful to Dr. Joseph Loo, Dr. Rachel Loo, and Dr. Anne G. Glaesener for critical feedback on the manuscript.

## Author contributions

**Conceptualization:** Sunnyjoy Dupuis, Usha F. Lingappa, Mary S. Lipton, Sabeeha S. Merchant.

**Data curation:** Sunnyjoy Dupuis, Samuel O. Purvine.

**Formal analysis:** Sunnyjoy Dupuis, Samuel O. Purvine, Lauren Chiang, Sean D. Gallaher.

**Funding acquisition:** Sunnyjoy Dupuis, Sabeeha S. Merchant.

**Investigation:** Sunnyjoy Dupuis, Usha F. Lingappa, Samuel O. Purvine, Carrie D. Nicora.

**Methodology:** Sunnyjoy Dupuis, Usha F. Lingappa, Samuel O. Purvine, Carrie D. Nicora.

**Project administration:** Sunnyjoy Dupuis, Sabeeha S. Merchant.

**Resources:** Mary S. Lipton, Sabeeha S. Merchant.

**Supervision:** Mary S. Lipton, Sabeeha S. Merchant.

**Validation:** Lauren Chiang.

**Visualization:** Sunnyjoy Dupuis, Lauren Chiang.

**Writing – original draft:** Sunnyjoy Dupuis, Samuel O. Purvine, Carrie D. Nicora.

**Writing – review & editing:** Sunnyjoy Dupuis, Usha F. Lingappa, Samuel O. Purvine, Lauren Chiang, Sean D. Gallaher, Carrie D. Nicora, Mary S. Lipton, Sabeeha S. Merchant.

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
