## [Decision Letter · Decision Letter 0]

4 Nov 2025

Mono-mix strategy enables comparative proteomics of a cross-kingdom microbial symbiosis

PLOS ONE

Dear Dr. Merchant,

Thank you for submitting your manuscript to PLOS ONE. After careful consideration, we feel that it has merit but does not fully meet PLOS ONE’s publication criteria as it currently stands. Therefore, we invite you to submit a revised version of the manuscript that addresses the points raised during the review process.

We look forward to receiving your revised manuscript.

Kind regards,

Rishiram Ramanan

Academic Editor

PLOS ONE

Journal Requirements:

[This work was supported by The Gordon and Betty Moore Foundation Symbiosis in Aquatic Systems Initiative Investigator Award GBMF9203 to S.S.M. (https://doi.org/10.37807/GBMF9203). Proteomic analysis was performed on a Facilities Integrating Collaborations for User Science (FICUS) project award to S.D. and S.S.M (10.46936/fics.proj.2021.60010/60000390) from the Environmental Molecular Sciences Laboratory, a Department of Energy Office of Science User Facility sponsored by the Biological and Environmental Research program under Contract No. DE-AC05-76RL01830. S.D. acknowledges support from National Institutes of Health T32 Genetic Dissection of Cells and Organisms training grant 1T32GM132022-01. U.F.L. acknowledges support from the µBiospheres Scientific Focus Area grant SCW1039 from the US Department of Energy Office of Biological and Environmental Research. The funders had no role in study design, data collection and analysis, decision to publish, or preparation of the manuscript.].

4. Thank you for stating the following in your manuscript:

[This work was supported by The Gordon and Betty Moore Foundation Symbiosis in Aquatic Systems Initiative Investigator Award GBMF9203 to S.S.M. (https://doi.org/10.37807/GBMF9203). Proteomic analysis was performed on a Facilities Integrating Collaborations for User Science (FICUS) project award  (10.46936/fics.proj.2021.60010/60000390) from the Environmental Molecular Sciences Laboratory, a Department of Energy Office of Science User Facility sponsored by the Biological and Environmental Research program under Contract No. DE-AC05-76RL01830. S.D. acknowledges support from the National Institutes of Health T32 Genetic Dissection of Cells and Organisms training grant 1T32GM132022-01. U.F.L. acknowledges support from the µBiospheres Scientific Focus Area grant SCW1039 supported by the Genome Sciences Program of the U.S. Department of Energy's Office of Biological and Environmental Research. The funders had no role in study design, data collection and analysis, decision to publish, or preparation of the manuscript.]

[This work was supported by The Gordon and Betty Moore Foundation Symbiosis in Aquatic Systems Initiative Investigator Award GBMF9203 to S.S.M. (https://doi.org/10.37807/GBMF9203). Proteomic analysis was performed on a Facilities Integrating Collaborations for User Science (FICUS) project award to S.D. and S.S.M (10.46936/fics.proj.2021.60010/60000390) from the Environmental Molecular Sciences Laboratory, a Department of Energy Office of Science User Facility sponsored by the Biological and Environmental Research program under Contract No. DE-AC05-76RL01830. S.D. acknowledges support from National Institutes of Health T32 Genetic Dissection of Cells and Organisms training grant 1T32GM132022-01. U.F.L. acknowledges support from the µBiospheres Scientific Focus Area grant SCW1039 from the US Department of Energy Office of Biological and Environmental Research. The funders had no role in study design, data collection and analysis, decision to publish, or preparation of the manuscript.]

6. PLOS requires an ORCID iD for the corresponding author in Editorial Manager on papers submitted after December 6th, 2016. Please ensure that you have an ORCID iD and that it is validated in Editorial Manager. To do this, go to ‘Update my Information’ (in the upper left-hand corner of the main menu), and click on the Fetch/Validate link next to the ORCID field. This will take you to the ORCID site and allow you to create a new iD or authenticate a pre-existing iD in Editorial Manager.

Additional Editor Comments:

It was difficult to secure reviewers for this manuscript as several reviewers rejected the invitation to review the manuscript. This is for your kind information.

Kindly execute the comments of the reviewers and submit a revised version at the earliest.

Reviewers' comments:

Reviewer's Responses to Questions

**Comments to the Author**

1. Is the manuscript technically sound, and do the data support the conclusions?

Reviewer #1: Yes

Reviewer #2: Yes

2. Has the statistical analysis been performed appropriately and rigorously?

Reviewer #1: Yes

Reviewer #2: Yes

3. Have the authors made all data underlying the findings in their manuscript fully available?

Reviewer #1: Yes

Reviewer #2: Yes

4. Is the manuscript presented in an intelligible fashion and written in standard English?

Reviewer #1: Yes

Reviewer #2: Yes

Reviewer #1: This manuscript is an excellent study that addresses a fundamental and important technical challenge in the proteomics analysis of cross-kingdom microbial symbioses, such as those between algae and bacteria. The problem of abundant host proteins masking the detection of proteins from a less abundant symbiont is a common and significant hurdle for researchers in this field. The authors are to be commended for clearly demonstrating this issue with data, while also presenting practical and effective solutions in the "mono-mix" strategy and sample fractionation. The experimental design is logical, the methodology is appropriate, and the conclusions are well-supported by the data. This work represents an important methodological contribution that will improve the reliability of dual-proteomics analyses of symbiotic systems and is well-suited for publication in PLOS ONE. The manuscript's quality would be further enhanced by addressing the minor revisions detailed below.

Minor Comments

To further improve the quality and clarity of the manuscript, I recommend the following minor revisions.

1. Emphasizing the limitations and practical recommendations for the "mono-mix" control

The "mono-mix" strategy is a creative core of this study. However, as the authors frankly acknowledge in the Discussion, a practical challenge is the difficulty in perfectly matching the biomass ratios between the coculture and the mono-mix control. This is the most critical limitation of the method. To ensure readers can implement this strategy without misinterpretation, it would be beneficial to more prominently highlight this limitation and the authors' proposed solutions (e.g., preparing multiple mono-mix ratios, freezing samples for post-hoc mixing ) in the Abstract or at the beginning of the Discussion.

2. Discussion of the protein quantification method

The study uses a "simple rollup" method (summing peptide signal intensities) for protein quantification. While this is a common approach, other quantification algorithms, such as MaxLFQ, are known to be more robust against outlier peptides. A single sentence in the Methods or Discussion briefly explaining the rationale for choosing this method, or acknowledging that other methods could also be applied, would further strengthen the methodological rigor of the paper.

3. Addition of a summary figure for the overall workflow

This is an excellent methods paper that contrasts a problematic approach with a proposed solution. I strongly recommend adding a schematic summary figure to help readers intuitively grasp the entire workflow. For instance, a figure that visually contrasts the "Conventional Path" (Monoculture vs. Coculture → False Positives) with the "Proposed Path" (Mono-mix + Fractionation vs. Coculture → True Biological Insight) would make the paper's contribution even more clear and impactful.

4. A note of caution for the interpretation of Figure 3 results

The PCA in Figure 3D shows that the continuous-light coculture samples (white triangles) are the most distinct from their mono-mix controls (white circles). While this is discussed as a biological response, the data in Figure 2B and 2C show that this same continuous-light condition was the one with the poorest match in biomass ratio between the coculture and the mono-mix. Therefore, the separation seen in the PCA could be influenced not only by biological changes but also by the residual "detection bias." A sentence should be added to the Discussion to acknowledge this as a potential confounding factor when interpreting these specific results.

Reviewer #2: This is a useful paper that addresses issues that are evident when working with proteomics with two different organisms, and in this case, quite contrasting (prokaryote and eukaryote). The experiments undertaken provide interesting insights into how quantitative information can be more reliable, leading to more confidence in biological interpretations. There are several aspects that need clarifying- such as the issues of imputation and also a more broad assessment of the current status of the field of co-culture proteomics. Specific comments have been added to the manuscript.

**Do you want your identity to be public for this peer review?** For information about this choice, including consent withdrawal, please see our Privacy Policy

Reviewer #1: No

Reviewer #2: No

---

## [Author Response · Author response to Decision Letter 1]

15 Dec 2025

As suggested, we have:

• Added a summary figure (R1 Fig 6), a graphic representation of the challenges that we addressed in the study and strategies that we tested for overcoming them.

• Added supplementary figure panels that demonstrate the p-values of the differential expression analyses.

• Emphasized our discussion of the limitations of the strategies explored in this work.

• Revised several sections of the Materials and Methods to more clearly describe our approaches to protein quantification and normalization.

These changes are detailed in the Point-by-Point Response to Reviewers file included with this submission. As requested by the Editors, we have included our amended Funding Statement in the cover letter included with the submission.

---

## [Editor Report · Decision Letter 1]

18 Dec 2025

Mono-mix strategy enables comparative proteomics of a cross-kingdom microbial symbiosis

PONE-D-25-40755R1

Dear Dr. Merchant,

We’re pleased to inform you that your manuscript has been judged scientifically suitable for publication and will be formally accepted for publication once it meets all outstanding technical requirements.

Kind regards,

Rishiram Ramanan

Academic Editor

PLOS One
---

## [Editor Report · Acceptance letter]

PONE-D-25-40755R1

PLOS One

Dear Dr. Merchant,

I'm pleased to inform you that your manuscript has been deemed suitable for publication in PLOS One. Congratulations! Your manuscript is now being handed over to our production team.

Kind regards,

on behalf of

Dr. Rishiram Ramanan

Academic Editor

PLOS One